# Layered van der Waals crystals with hyperbolic light dispersion

M.N. Gjerding[1,2], R. Petersen[3,4], T.G. Pedersen[3,4], N.A. Mortensen [2,5,6] & K.S. Thygesen [1,2]

Compared to artificially structured hyperbolic metamaterials, whose performance is limited by the finite size of the metallic components, the sparse number of naturally hyperbolic materials recently discovered are promising candidates for the next generation of hyperbolic materials. Using first-principles calculations, we extend the number of known naturally hyperbolic materials to the broad class of layered transition metal dichalcogenides (TMDs). The diverse electronic properties of the transition metal dichalcogenides result in a large variation of the hyperbolic frequency regimes ranging from the near-infrared to the ultra-violet. Combined with the emerging field of van der Waals heterostructuring, we demonstrate how the hyperbolic properties can be further controlled by stacking different two-dimensional crystals opening new perspectives for atomic-scale design of photonic metamaterials. As an application, we identify candidates for Purcell factor control of emission from diamond nitrogen-vacancy centers.

[1] CAMD, Department of Physics, Technical University of Denmark, Kongens Lyngby 2800, Denmark. [2] Center for Nanostructured Graphene (CNG), Technical University of Denmark, Kongens Lyngby 2800, Denmark. [3] Department of Physics and Nanotechnology, Aalborg University, Aalborg East 9220, Denmark. [4] Center for Nanostructured Graphene (CNG), Aalborg East 9220, Denmark. [5] Center for Nano Optics, University of Southern Denmark, Campusvej 55, Odense M 5230, Denmark. [6] Department of Photonics Engineering, Technical University of Denmark, Kongens Lyngby 2800, Denmark. Correspondence and requests for materials should be addressed to K.S.T. (email: thygesen@fysik.dtu.dk)

T he dispersion relation of light in a homogeneous, anisotropic-layered medium is determined by the equation

$$\frac{k_\parallel^2}{\varepsilon_\perp(\omega)} + \frac{k_\perp^2}{\varepsilon_\parallel(\omega)} = \frac{\omega^2}{c^2}, \qquad (1)$$

where $\parallel$ and $\perp$, respectively, indicate the in-plane and out-of-plane components of the dielectric function, $\varepsilon$, and the wave vector, $k$. For frequencies where the in-plane and out-of-plane dielectric functions differ in sign, the isofrequency surfaces become hyperbolic with an apparent divergence of the photonic density of states. Effective medium theory[1] (EMT) predicts that metamaterials composed of metallic nanostructures embedded in a dielectric medium, as sketched in Fig. 1a, b, become hyperbolic over wide frequency ranges and that effects, such as the enhanced Purcell factor, become broadband[2–4], in stark contrast to alternative methods for Purcell factor enhancement based on resonant cavities or localized surface plasmon resonances that are inherently narrow band[5]. Additional unusual properties include negative refraction and hyperlensing[6], also originating from the sub-wavelength structured anisotropy. Due to their relatively simple structural components and multitude of interesting optical properties, artificially structured materials with hyperbolic light dispersion therefore represent one of the most intriguing and useful types of electromagnetic metamaterials[1, 7, 8].

In reality, however, EMT and the homogeneity assumption underlying Eq. (1) break down when the electromagnetic modes vary on a length scale comparable to the structural periodicity of the material ($d$)[9]. Consequently, the hyperbolic dispersion of the medium becomes limited to wave vectors smaller than $k_{max} \sim \pi/d$, see Fig. 1. On the microscopic level, the broadband response of the metamaterial results from the hybridization of surface plasmons at the metal-dielectric interfaces. This implies that the broadband response can be realized only if the surface plasmons have significant spatial overlap over a wide frequency range. It turns out that this condition is hardly fulfilled for conventional metamaterials[10]. In principle, the problem could be overcome by reducing the film thickness. However, downscaling of metal-dielectric structures to sub-10 nm periodicity is highly challenging and inevitably leads to increased interface roughness and enhanced scattering losses that are detrimental to metamaterial performance[11].

Recently, a few materials have been reported to exhibit hyperbolic dispersion in their pristine form[12–14]. Hexagonal boron nitride was shown to be hyperbolic in its reststrahlen band between the longitudinal and transverse optical phonon branches[15, 16], while graphite[12] and the tetradymites $Bi_2Se_3$ and $Bi_2Te_3$[13] are hyperbolic in the UV and near-IR to visible, respectively. Natural hyperbolic materials have the obvious advantages over traditional metamaterials that they require no artificial structuring and contain no internal interfaces for the electrons to scatter off. Moreover, due to their homogeneous nature, the hyperbolic dispersion is expected to extend much further in reciprocal space (Fig. 1c)—in principle-only limited by the atomic periodicity of the crystal lattice or effects of non-local dielectric response. Although these advantages appear plausible their significance have so far not been quantified.

Here we report on the discovery of a large class of natural hyperbolic materials with optical properties greatly surpassing those of conventional metamaterials and presently known natural hyperbolic materials. Out of a set of 31 layered transition metal dichalcogenides (TMDs), our first-principles calculations predict hyperbolic dispersion for all the materials over a broad frequency span from the near-IR to the UV. As a unique feature, compared to presently known natural hyperbolic materials which all are hyperbolic only above the onset of interband transitions,

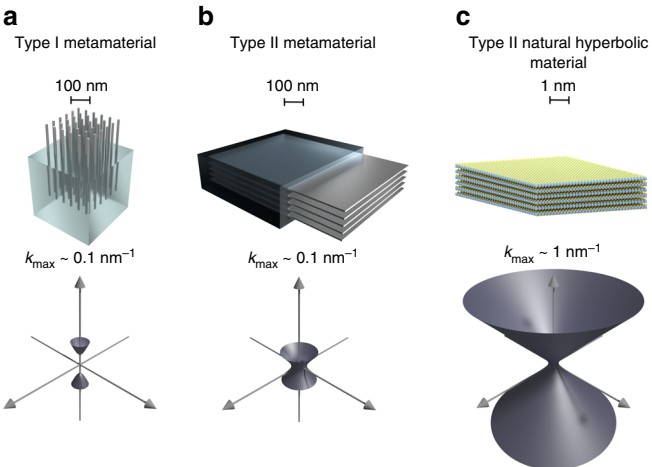

**Fig. 1** Hyperbolic materials. Metal nanowire (**a**) and planar multilayer (**b**) metamaterials are artificially structured materials engineered to exhibit metallic response in some directions and dielectric response in others. In the effective medium limit, this anisotropy leads to hyperbolic isofrequency surfaces, which in principle would result in a diverging photonic density of states. In reality, the hyperbolic dispersion is only realized for wave vectors up to $k_{max} \sim \pi/d$, where $d$ is the periodicity of the medium which is typically on the order of tens of nanometers. Natural hyperbolic materials (**c**) possess no artificial structuring and their hyperbolic dispersion extends much further in reciprocal space being limited only by the atomic periodicity of the crystal structure

the metallic TMDs can be hyperbolic below the interband threshold leading to extremely large broadband Purcell factors governed by weakly damped hyperbolic modes. Inspired by experimental demonstrations that van der Waals (vdW) hetero-structures can be constructuted by stacking different TMDs[17] we show, using EMT to evaluate dielectric tensors, how the hyperbolic properties can be further tuned by heterostructuring allowing for the design of atomic-scale structured metamaterials with Purcell factors tailored for specific emission sources.

## Results

**Naturally hyperbolic transition metal dichalcogenides.** We used first-principles density functional theory (DFT) and linear response calculations to investigate the optical properties of 31 non-magnetic TMDs with experimentally known crystal structures[18]. Figure 2 shows the calculated hyperbolic spectral range of the TMDs. Quite surprisingly all the materials, including metals (in the sense of having a partially filled conduction band) and semiconductors, are predicted to be hyperbolic in some frequency range. The materials exhibit predominantly type II dispersion corresponding to metallic response in-plane and dielectric response out-of-plane. The strong dielectric anisotropy arises from the weak interlayer coupling that lowers carrier velocities and plasma frequencies perpendicular to the layers. We note that while the qualitative trends predicted by our DFT calculations, that employ the Perdew–Burke–Ernzerhof (PBE) exchange-correlation (xc) functional, are very reliable, the quantitative accuracy is subject to some uncertainty. In particular, PBE is known to underestimate band gaps and interband transition energies, which is expected to influence the optical properties including the precise position of the hyperbolic regimes. To investigate the role of the xc-functional, we have compared the dielectric function of $2H$-$TaS_2$ obtained with the PBE and the more accurate but computationally expensive HSE[19] functional (see Supplementary Fig. 1). We find that the HSE blueshifts the interband transitions by around 0.5 eV relative to the PBE result.

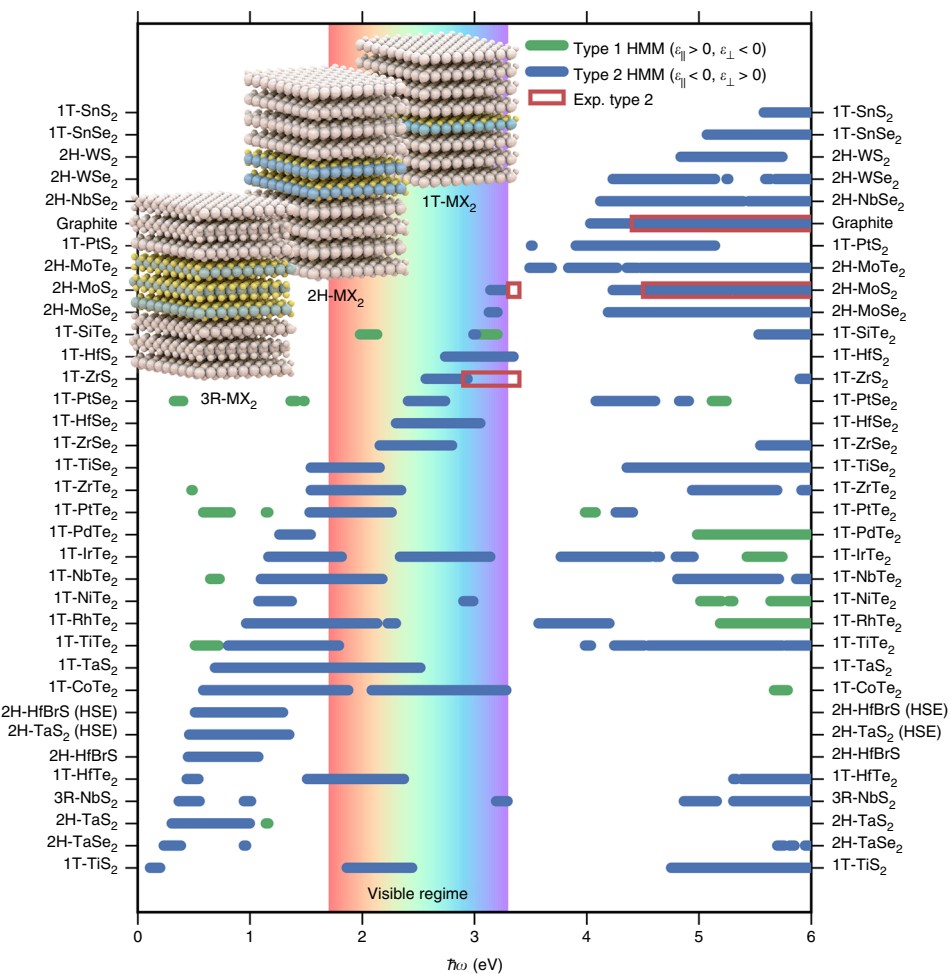

**Fig. 2** Calculated hyperbolic regimes of the TMDs and graphite. The hyperbolic frequency ranges of the transition metal dichalcogenides (*TMDs*) investigated in this work. The hyperbolic dispersion of the TMDs is predominantly Type II (*blue*) corresponding to metallic response in-plane and dielectric response out-of-plane. The dielectric functions have been calculated on top of DFT-PBE wave functions. The result for 2H-TaS$_2$ and 2H-HfBrS obtained with the HSE functional is shown for comparison. Experimental hyperbolic regimes were derived from experimental in-plane and out-of-plane dielectric functions reported in: Graphite[20], 2H-MoS$_2$[21, 22], and 1T-ZrS$_2$[23]

These shifts change the type II hyperbolic regime of 2H-TaS$_2$ from 0.3–1.1 eV (PBE) to 0.4–1.3 eV (HSE), see Fig. 2. Similar changes are expected for the other materials.

We compare our results to the hyperbolic regimes of the few layered compounds for which we have found experimental data for both the in-plane and out-of-plane dielectric functions: Graphite[20], 2H-MoS$_2$[21, 22], and 1T-ZrS$_2$[23]. The experimental hyperbolic regimes are marked in Fig. 2 by *red squares* and are in good agreement with our predictions. Unfortunately, other studies of the optical properties of the TMDs probe only the in-plane dielectric response[21, 24–27] from which it is not possible to extract the experimental hyperbolic regimes. Therefore, to extend our comparison with experiments, we have also compared our predictions to the experimental regimes in which the in-plane dielectric function is negative which is typically a good indicator for type 2 hyperbolicity (Supplementary Fig. 5). This comparison shows that the qualitative trends of the predicted hyperbolic regimes are well described and that the quantitative accuracy is on the order of 0.5 eV compared to the experimental results. Furthermore, compounds with multiple experimental studies show significant variations in the experimentally determined hyperbolic regimes (based on the $\varepsilon_\parallel < 0$ criterion). For example, different experiments on 1T-TiSe$_2$ predict hyperbolic regimes of 0.9–2.3 eV[25] and 1.5–2.2 eV[26] and experiments on 2H-MoS$_2$

predict hyperbolic regimes from 3.2 to 3.3 eV[21], none at all[22] and measurements on three layers indicate a lower limit on the order of 2.9 eV[28]. A comparison between the calculated dielectric properties and the experimental data can be found in the Supplementary Information (Supplementary Figs. 7–39). Finally, with studies reporting growth on the centimeter scale[29, 30], we do not expect any technical difficulties in the experimental synthesis of high-quality materials with large lateral areas.

**Giant Purcell factor.** We compare the performance of some of the newly discovered natural hyperbolic materials to graphite, a well-known hyperbolic material[12], and judge their performance on the basis of the calculated Purcell factor for a dipole placed 1 nm above a bulk substrate. The Purcell factor is defined as the ratio between the actual decay rate of the emitter ($\Gamma$) and the free-space value ($\Gamma_0$) and was calculated as described in the "Methods" section. The results are shown in Fig. 3a for two representative TMDs: the metallic 2H-TaS$_2$, hyperbolic from 0.4 to 1.3 eV (we use the HSE result for this example), and the semiconducting 1T-ZrS$_2$, hyperbolic from 2.5 to 2.8 eV. The hyperbolic regimes from Fig. 2 are marked by *broadened lines*. For graphite and 1T-ZrS$_2$, our results are in excellent agreement with the calculated Purcell factors based on experimental dielectric functions[20, 23]

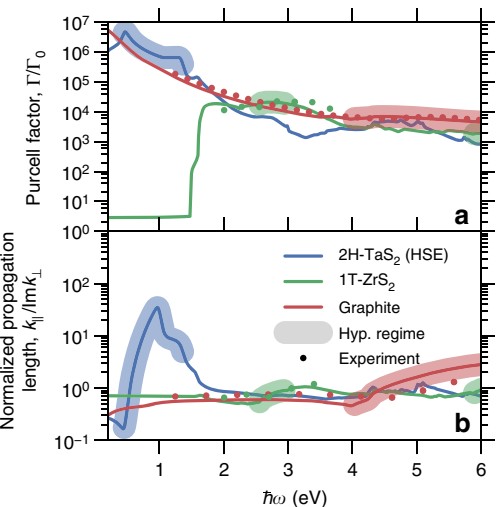

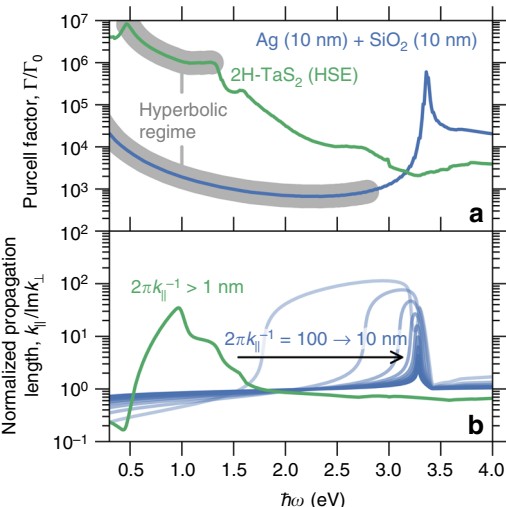

**Fig. 3** Purcell factors for metallic and semiconducting natural hyperbolic materials. **a** Purcell factor for a dipole placed a distance of 1 nm above the substrate and **b** propagation lengths of hyperbolic modes in direction normal to the substrate normalized by the in-plane wavelength. *Lines* indicate results based on first-principles calculations and *dots* indicate the results obtained from the experimental dielectric functions[20, 23]. The *broadened patches* on each *line* indicate the calculated hyperbolic regimes of Fig. 2

**Fig. 4** Natural hyperbolic materials vs. hyperbolic metamaterials. **a** Purcell factors and **b** propagation lengths for a 50 nm thick film of the TMD 2H-TaS$_2$ compared with a 100 nm thick hyperbolic metamaterial consisting of alternating Ag-SiO$_2$ films with a period of 20 nm (10 nm Ag—10 nm SiO$_2$). The dipole is placed a distance of $h = 1$ nm above the substrate. The Purcell factors were calculated using the transfer matrix method (*full lines*). The propagation lengths are shown for various in-plane wave vector components corresponding to wavelengths in the nanometer regime $\left(\lambda = 2\pi k_{\parallel}^{-1}\right)$

(points). Our results show that 2H-TaS$_2$ exhibits giant Purcell factors on the order of $10^7$ in the technologically important near infrared (NIR) spectral range, while the semiconducting 1T-ZrS$_2$ offers large Purcell factors only for frequencies above its band gap of ~1.5 eV. We find similar results for the other metallic and semiconducting TMDs (Supplementary Fig. 4). It is important to note that the large Purcell factor of graphite in the NIR is caused by its large optical losses (quenching) rather than hyperbolic modes as is the case for 2H-TaS$_2$.

The metallic nature of 2H-TaS$_2$ is key to its simultaneously large Purcell factor and low losses. Due to the dramatic increase of damping above the onset of interband transitions, the Purcell factor of semiconductors is completely dominated by quenching and the normalized propagation length of hyperbolic modes (in the direction normal to the substrate, normalized by the in-plane wavelength), Fig. 3b, rarely exceeds unity. Our calculations show that quenching is severe for graphite and we expect similar results for all other known natural hyperbolic materials none of which are metals. In contrast, clear signatures on the Purcell factor enhancement due to propagating hyperbolic modes are observed for the metallic TMDs showing that quenching does not dominate the properties of the metals, as anticipated. The long propagation lengths of the hyperbolic modes in 2H-TaS$_2$ is a consequence of its special band structure with conduction bands being separated from other bands by finite energy gaps, which reduces damping at finite frequencies[31]. The effect of the special bandstructure has been modeled using a simple model for the Drude scattering rate that takes into account electron-hole pair excitations (see ref. [31] for details). These results clearly demonstrate the superior properties of the metallic hyperbolic materials for the spontaneous emission enhancement and propagation and as a final remark we note that TaS$_2$ shows particularly good properties at the telecom wavelength (1.55 μm or 0.8 eV) exhibiting simultaneously large propagation lengths and Purcell factors.

The giant Purcell factors exhibited by the natural hyperbolic materials are caused by their lack of internal structure. To show this, we compare the performance of the natural hyperbolic

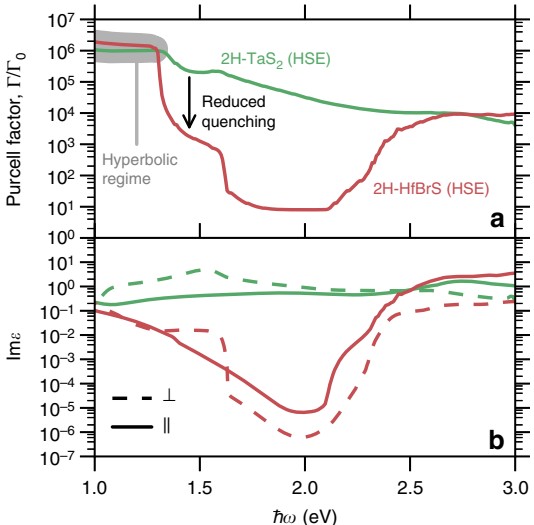

**Fig. 5** Reduced quenching of spontaneous emission. Reduced optical losses in 2H-HfBrS compared to 2H-TaS$_2$ significantly reduces the associated quenching of spontaneous emission, which is evident when comparing **a** Purcell factors of 50 nm thin films with **b** the imaginary part of their dielectric functions

TMDs to a conventional metamaterial using the transfer matrix method to calculate the Purcell factor for a point dipole placed 1 nm above the surface of 2H-TaS$_2$ and a Ag-SiO$_2$ heterostructure, respectively (see Fig. 4a). 2H-TaS$_2$ was chosen due to its interesting metallic electronic properties. For the Ag-SiO$_2$ structure, we used a fill-fraction of 50% and a periodicity of $d = 20$ nm. The Ag-SiO$_2$ structure is hyperbolic in the range 0–2.8 eV.

The broadband Purcell factor of TaS$_2$ exceeds that of the metamaterial by 2–3 orders of magnitude. This is a direct

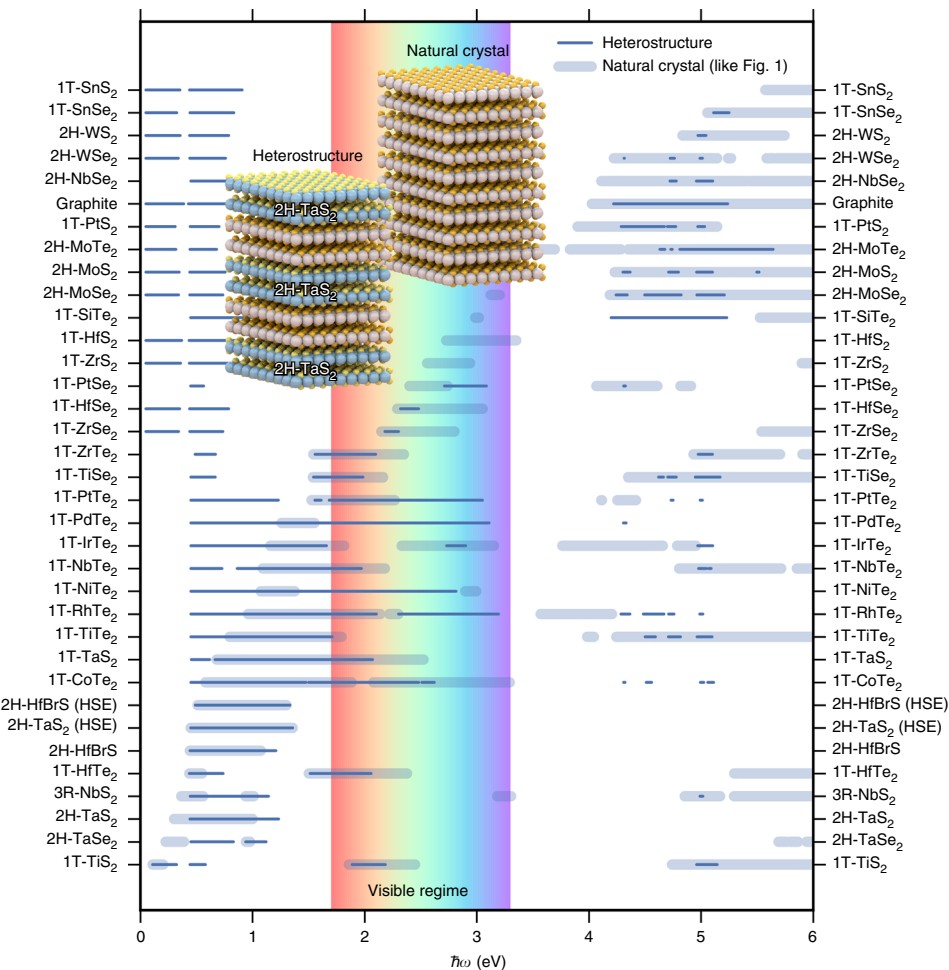

**Fig. 6** Hyperbolic van der Waals heterostructures. The hyperbolic spectral regimes of heterostructures consisting of 50% 2H-TaS$_2$ and 50% of another TMD (*narrow lines*). For comparison, the hyperbolic regimes of the pristine TMDs are repeated from Fig. 1 (*faded wide lines*). The hyperbolic regimes of the heterostructures can differ substantially from those of the constituent compounds

consequence of the homogeneous nature of the TaS$_2$ crystal, which extends the hyperbolic isofrequency surface to atomic-scale wavevectors. Indeed, in the present example, the Purcell factor is not limited by the intrinsic properties of the TaS$_2$ crystal but rather by the finite distance between the emitter and the surface, which introduces a natural cutoff on the wave vectors that can couple to the near-field of the emitter. For distances below 1 nm, the Purcell factor of TaS$_2$ increases further while that of the Ag-SiO$_2$ metamaterial remains constant being limited by the heterostructure periodicity. When the emitter is brought closer to the surface, the Purcell factor of TaS$_2$ is eventually limited by the non-local response of the medium (also known as spatial dispersion)[32], which arises from the atomic periodicity of the crystal and manifests itself in a non-trivial wavevector dependence of the dielectric function[33]. A detailed analysis of the effect of non-local response is provided in the Supplementary Note 1 and Supplementary Fig. 2. The relatively low Purcell factor of the Ag-SiO$_2$ structure is due to the small hyperbolic isofrequency surface and corresponding small enhancement of the density of states. The strong decay of the electromagnetic field inside the Ag layers leads to negligible hybridization between the surface plasmons implying that the hyperbolic response is lost (as evidenced by comparing to the Purcell factor predicted by EMT, Supplementary Fig. 6). This explains the narrow peak in the Purcell factor at 3.3 eV originating from the unhybridized Ag surface plasmon.

Applications such as hyperlensing depend critically on the propagation of light modes through the hyperbolic medium (Fig. 4b). The relatively low-propagation lengths found for the Ag-SiO$_2$ metamaterial originate from the weak hybridization of the silver surface plasmons, which leads to weak dispersion and thus low group velocities in the normal direction. The attenuation length, $\kappa^{-1}$, of the electromagnetic field inside the Ag slabs is determined by the dispersion $\kappa^2 = k_{\parallel}^2 - \epsilon_{Ag}(\omega)\omega^2/c^2$[34] explaining the observed dependence of the propagation length on frequency and in-plane wavevector. In comparison, the hyperbolic modes in TaS$_2$ propagate deeper into the film and exhibit no dependence on wavelength.

**Spontaneous emission on demand.** Materials exhibiting a large variation in the Purcell factor over a narrow frequency range could be used to control the emission rate of an emitter via the emission frequency[35]. Such control is essential for generating single photons on demand as required by many applications in quantum information technology[36, 37]. Returning to Fig. 4a, we note that 2H-TaS$_2$ shows a rather modest decrease in Purcell factor outside the hyperbolic regime (0.4–1.3 eV). The reason for the weak relative change in Purcell factor is quenching, i.e., enhanced emission due to evanescent modes existing at the surface of the material both inside and outside the hyperbolic frequency range. The contribution to the Purcell factor from such

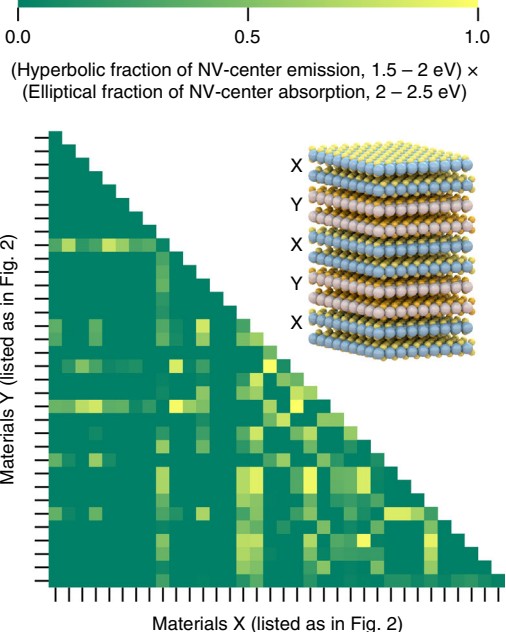

**Fig. 7** Heterostructures with tailored hyperbolic dispersion. The possibility of tuning the hyperbolic dispersion by combining different two-dimensional crystals into van der Waals heterostructures is illustrated by the search for a two-component TMD superlattice with hyperbolic (normal) dispersion in the emission (absorption) spectrum of an NV-color center in *diamond*. The *color scheme* indicates the fraction of hyperbolic to normal dispersion over the emission band (1.5–2 eV) times the fraction of normal to hyperbolic dispersion over the absorption band (2–2.5 eV). *Lighter color* thus corresponds to better candidate materials. The sheer number of good candidates found indicates that significant control of the optical properties can be achieved via vdW heterostructuring

evanescent modes is proportional to the imaginary part of the dielectric constant[2], suggesting that Purcell factors with large on/off ratios may be realized in hyperbolic materials with small $Im\varepsilon$ throughout the hyperbolic frequency regime. The chalcogen-halogen mixed compound 2H-HfBrS, which was recently identified by computational screening of low-loss materials for plasmonics[31], represents such a material. The electronic band structure of 2H-HfBrS features a single metallic band that is separated from all higher and lower lying bands by finite energy gaps. This type of band structure suppresses the number of final states for scattering in an energy range between the intraband and interband transitions, leading to a small imaginary part of the dielectric function. Figure 5a compares the calculated Purcell factors of 2H-TaS$_2$ and 2H-HfBrS. It is clear that 2H-HfBrS shows a much more dramatic decrease in Purcell factor when the frequency is moved outside the hyperbolic regime. The *lower panel* of the figure (Fig. 5b), showing the imaginary part of the dielectric functions, reveals that this difference is indeed due to the lower losses in 2H-HfBrS, which suppress the effect of quenching on the Purcell factor.

**Tuning by heterostructuring**. Next, we have explored the possibility of controlling light dispersion by combining layers of different TMDs into vdW heterostructures[17]. The scientific field of vdW heterostructuring represents an exciting new avenue for the nano-scale tuning of electronic and optical properties[17] and has already lead to numerous practical realization of nano-scale engineered devices (e.g., field effect tunneling transistors, photo-detectors, and light emitting diodes). The practical feasibility of

the proposed large-scale nanostructured heterostructures is supported by an increasing amount of experimental studies showing practical realizations of high-quality large area heterostructures[38–42] (on the order of 10 μm).

For hyperbolic metamaterials, an equivalent degree of control has traditionally been provided by varying the fill-fraction or by changing the metal or dielectric components of the hyperbolic metamaterial[43]. However, in order to ensure sufficient hybridization between surface plasmons at the interfaces, the metallic fill-fraction cannot be too large (the example discussed in relation to Fig. 4 shows that this problem is severe even for metal films as thin as 10 nm). Moreover, the number of useful metal-dielectric combinations is limited by the lattice matching required when fabricating high-quality thin films. In contrast, the weak interlayer bonding in van der Waals crystals relaxes the requirement of lattice matching at the interfaces thereby removing the restriction on the type of materials that can be combined. Furthermore, vdW interfaces can be atomically sharp presenting much fewer defects than covalently bonded, epitaxially grown interfaces[17].

To illustrate the degree of tuning that can be achieved by vdW heterostructuring, we have used EMT to calculate the hyperbolic frequency regimes of heterostructures obtained by combining each of the investigated TMDs with 2H-TaS$_2$ (50% fill-fraction). By employing EMT, we neglect the influence of hybridization at the interfaces and quantum confinement, which we have justified by first-principles calculations showing that the hyperbolic regimes do not depend significantly on either of these effects (see Supplementary Note 2). Figure 6 shows that the process of heterostructuring can shift, expand, or reduce the hyperbolic frequency regime as compared to those of the constituent materials.

As a specific design example, suppose we are interested in a material with hyperbolic dispersion in the spectral range 1.5–2 eV and normal (elliptical) dispersion in the spectral range 2–2.5 eV. The two intervals correspond, respectively, to the emission and absorption spectral range of a nitrogen-vacancy (NV) center in diamond[44], but this is not essential for the present proof-of-concept illustration. We have calculated the hyperbolic regimes of heterostructures formed by combining any two TMDs with a 50% filling fraction. Figure 7 shows the fraction of the interval 1.5–2 eV, over which the heterostructure dispersion is hyperbolic, times the fraction of the interval 2–2.5 eV where it is elliptical. As can be seen several heterostructure candidates are discovered. The fact that the best candidates are not found on the diagonal (corresponding to the pristine TMDs) shows that the process of vdW heterostructuring represents a genuine extension of the space of natural hyperbolic materials. Further tuning is obviously possible by varying the fill-fraction or by considering heterostructures with more than two material components.

## Discussion

Our work not only identifies a new class of natural hyperbolic materials, with the metallic compounds being especially suitable for low-loss applications, but suggests that many other layered compounds may exhibit similar properties. We expect that these findings will stimulate further work and open new avenues for photonic metamaterials.

## Methods

**First-principles calculations**. All DFT calculations were performed using the GPAW electronic structure code[45] using a plane-wave basis with energy cutoff of 600 eV. The dielectric functions of the TMDs were calculated within the random-phase approximation using single-particle wave functions and energies obtained from DFT calculations employing the PBE exchange-correlation (xc) functional[46]. Brillouin zone integrals were performed over a k-point grid of density of 30 Å⁻¹ using the linear tetrahedron method[47]. Unoccupied bands up to at least 40 eV

above the Fermi energy were included in the band summation. Local-field effects were included up to a plane-wave cutoff of 60 eV.

To account for higher-order scattering processes, such as phonon- or defect-mediated intraband transitions, we include a phenomenological relaxation rate of the form $\gamma(\omega) = a\mathrm{JDOS}(\omega)/\omega$, where $\mathrm{JDOS}(\omega)$ is the joint density of states per volume and $a$ is a parameter controlling the scattering strength[31] $a$ is highly material- and sample specific and depends on defect concentration, defect type, electron-phonon coupling strength, etc. For consistency, we use the same value of $a$ for all materials considered in this work.

**HSE calculations**. HSE band structure calculations were performed for 2H-TaS$_2$ and 2H-HfBrS. In both cases, the primary effect is to increase the interband transitions by 0.5 eV compared to the PBE result. The HSE response was calculated based on scissor-corrected PBE eigenvalues and matrix elements were calculated using the PBE wave functions.

**Purcell factor**. The Purcell factor of a point dipole in the vicinity of a semi-infinite planar substrate can be derived from the knowledge of the substrate's Fresnell reflection coefficients and is given by[48]

$$\frac{\Gamma}{\Gamma_0} = 1 + \frac{3}{2k_c}\mathrm{Re}\left(\int_0^\infty k_\parallel \frac{\mathrm{d}k_\parallel}{k_{\perp,c}}\left[f_\perp^2 \frac{k_\parallel^2}{k_c^2}r_\mathrm{p} + \frac{1}{2}f_\parallel^2\left(r_s - \frac{k_{\perp,c}^2}{k_c^2}r_\mathrm{p}\right)\right]e^{2ik_\perp h}\right) \quad (2)$$

where $k_c = k_0\sqrt{\varepsilon_c}$, $k_{\perp,c} = \sqrt{k_c^2 - k_\parallel^2}$, $f_\perp^2 = |\mu \cdot \mathbf{z}|^2/(\mu^* \cdot \mu)$, $f_\parallel^2 = 1 - f_\perp^2$, $\varepsilon_c$ is the dielectric constant of the cladding, $\mu$ is an electric dipole placed a distance $h$ above the material, and $r_\mathrm{p}$ ($r_s$) is the Fresnell reflection coefficient for p (s)-polarized light. The reflection coefficients are given by

$$r_s = \frac{k_\perp - k_{\perp,\mathrm{m}}^s}{k_\perp + k_{\perp,\mathrm{m}}^s}, \quad r_\mathrm{p} = \frac{k_\perp \varepsilon_\mathrm{x} - k_{\perp,\mathrm{m}}^\mathrm{p}\varepsilon_c}{k_\perp \varepsilon_\mathrm{x} + k_{\perp,\mathrm{m}}^\mathrm{p}\varepsilon_c}, \quad (3)$$

where

$$k_{\perp,\mathrm{m}}^s = \sqrt{k_0^2\varepsilon_\parallel - k_\parallel^2}, \quad k_{\perp,\mathrm{m}}^\mathrm{p} = \sqrt{k_0^2\varepsilon_\parallel - \frac{\varepsilon_\parallel}{\varepsilon_\perp}k_\parallel^2} \quad (4)$$

and $k_0 = \omega/c$ is the free space wave vector.

**Propagation lengths**. The propagation length of the Ag-SiO$_2$ heterostructure was calculated for an infinite periodic structure. The periodicity allows the definition of a Bloch wave vector perpendicular to the interfaces $k_\mathrm{B}$. The propagation length through the slab is then $L = (\mathrm{Im}k_\mathrm{B})^{-1}$, which we normalize by the in-plane wavevector $k_\parallel$. The Bloch momentum is obtained by solving the well-known dispersion relation for planar periodic media[48],

$$\cos[k_\mathrm{B}(d_\mathrm{m} + d_\mathrm{d})] = \cos(k_{\perp,\mathrm{m}}d_\mathrm{m})\cos(k_{\perp,\mathrm{d}}d_\mathrm{d})$$
$$- \frac{1}{2}\left(\frac{\varepsilon_\mathrm{m}k_{\perp,\mathrm{d}}}{\varepsilon_\mathrm{d}k_{\perp,\mathrm{m}}} + \frac{\varepsilon_\mathrm{d}k_{\perp,\mathrm{m}}}{\varepsilon_\mathrm{m}k_{\perp,\mathrm{d}}}\right)\sin(k_{\perp,\mathrm{m}}d_\mathrm{m})\sin(k_{\perp,\mathrm{d}}d_\mathrm{d}), \quad (5)$$

where $d_\mathrm{m}$ and $d_\mathrm{d}$ are the metal and dielectric layer thicknesses, respectively, $k_{\perp,\mathrm{m}} = \sqrt{k_0^2\varepsilon_\mathrm{m} - k_\parallel^2}$ and $k_{\perp,\mathrm{d}} = \sqrt{k_0^2\varepsilon_\mathrm{d} - k_\parallel^2}$ are the transverse wave numbers for the metal and dielectric, respectively, and $\varepsilon_\mathrm{m}$ and $\varepsilon_\mathrm{d}$ are the dielectric functions of the metal and dielectric components, respectively.

**Data availability**. Calculated Purcell factors and propagation lengths are available from the corresponding author on reasonable request. The authors declare that all other data supporting the findings of this study are available within the paper and its Supplementary Information files.

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

## Acknowledgements

The Center for Nanostructured Graphene (CNG) is sponsored by the Danish National Research Foundation, Project No. DNRF103. T.G.P. was supported by VKR center QUSCOPE. N.A.M. is a VILLUM investigator supported by VILLUM Fonden. The project received funding from the European Unions Horizon 2020 research and innovation program under grant agreement no. 676580 with The Novel Materials Discovery (NOMAD) Laboratory, a European Center of Excellence.

## Author contributions

M.N.G. performed the DFT linear response calculations and R.P. calculated the Purcell factors. K.S.T. and T.G.P. supervised the project. All authors contributed to the discussion and data analysis. The first draft was written by M.N.G. and all authors commented on the manuscript.

## Additional information

**Competing interests:** The authors declare no competing financial interests.

