## [Peer Review File · Nature Communications]

Reviewers' Comments:

Reviewer #1 (Remarks to the Author)

Hyperbolic metamaterials has been considered as one of the most practical optical metamaterials and have drawn a lot of attentions. Recently, people found out there are natural existing crystals expressing hyperbolic dispersion, unless the optical loss is still quite high, especially at visible light frequencies. This manuscript studies the hyperbolic material properties based on layered transition metal dichalcogenides and shows the loss might be less than artificially structured metamaterials. The results seems to be very interesting. However, the reviewer do have a few questions regarding the modeling validity.

1. For the loss comparison between 2H-TaS₂ and Ag/SiO₂ multilayer in Fig. 3, the transmission peaks for these two kind of materials are different. The authors should find two systems have the same transmission peak and compare to each other.
2. It will be better if the authors can also compare the loss properties by including the case shown in Fig. 1(a).
3. The optical properties of the heterostructures was calculated by the EMT (equation S3 and S4). This may not be correct if the material properties of each atomic layer components was just set to be the same as its bulky value. Does quantum effect play an important role here? This have to be explained in great details, perhaps in supporting materials.

Reviewer #2 (Remarks to the Author)

While current manuscript considerably extends the number of known natural hyperbolic metamaterials, I do not find these findings novel enough. It is known for at least five years that several high T_c superconducting metamaterials are naturally hyperbolic. There was also a recent review of natural hyperbolic metamaterials published by Nature Photonics (E. E. Narimanov and A. V. Kildishev, "Metamaterials: Naturally hyperbolic", Nature Photonics 9, 214–216 (2015)). It is well understood that many materials having "layered" crystalline structure are hyperbolic. Thus, main finding of this manuscript is not new.

The Authors also calculate Purcell factors for these materials using a well established formalism. However, they do not report any experimental data, which would confirm their findings. I cannot recommend this manuscript for publication in Nature Communications.

Reviewer #3 (Remarks to the Author)

The authors report the discovery of a class of non-magnetic transition metal dichalcogenide (TMD) compounds that have strong optical anisotropy exhibiting hyperbolic dispersion in the optical spectrum, using first-principles density functional theory (DFT) calculations. The authors then go on to calculate the Purcell factor for spontaneous emission enhancement in these compounds and show that it is much higher than that of a Ag/SiO₂ multilayered hyperbolic metamaterial (HMM). Then, they show that by combining different types of TMD HMMs in a layered fashion using van der Waal's forces, one can tailor further the range of hyperbolic dispersion.

Considering that conventional HMMs are limited by losses and lower propagation lengths, I find this work, very intriguing and possibly as a fresh direction for the field of HMM.

However, I am of the opinion that it is not suitable for publication in a journal like Nature Communications at this stage.

The main reason is that it is fully theoretical and there is no comparison with any experimental findings to validate the results shown here. There is a recent publication ["Imaging spectroscopic ellipsometry of MoS₂," J. Phys.: Condens. Matter 28(2016) 385301] that has shown ellipsometry measurements on MoS₂ (single and multilayer) and used an anisotropic model to describe its optical response. This paper, in fact, shows Type- II hyperbolic dispersion in the high energy (~3 eV) range. While, this is qualitatively similar to what the authors predict in this paper, it would considerably strengthen the results if they can also do a quantitative comparison of the dielectric constants provided in the above paper with those from their calculations and discuss the comparison of parameters such as Purcell factor and propagation length for the experimental and DFT dielectric constants. This would address some of the practical considerations too. Also, I suggest this publication should be cited by the authors as it has essentially demonstrated hyperbolic dispersion in MoS₂.

In addition to this, I have the following questions/comments:

- 1) Main paper, Page 4, paragraph 2 – The authors write: "As a unique feature, the metallic TMDs are hyperbolic below the onset of interband transitions....". Does this imply that the hyperbolic dispersion feature arises from intraband transitions? If so, how is it that DFT calculations can take into account such transitions? Secondly, if interband transitions have no role to play in the hyperbolic property, does it imply that the particular crystal structure of these materials is not a criterion in determining the hyperbolic dispersion behaviour?
- 2) As a follow up question, the authors refer to the TMDs as "metallic". I am curious in what respect they are metallic? They have metal-like dielectric constants (with negative real part) in one of the directions (or one plane) over a certain frequency range. But, in the orthogonal direction, the dielectric constant is not metal-like. So, I am wondering how the TMDs can be referred to as metallic? Is this from the perspective of DC conductivity/resistivity where they show properties similar to those of metals?
- 3) Are there any practical difficulties in realizing such HMMs, for example, say, ensuring uniformity of behavior over a large lateral area (both with pristine TMDs and those from van der Waals heterostructuring)?
- 4) Supplementary section S2 – What was the value for the electron relaxation rate used in the calculations of the Drude response (equations S1 and S2)? How was this determined?
- 5) Supplementary figure S2(a) – What are the two different cases (dotted and solid lines) of dielectric constants plotted? This does not seem to be mentioned clearly in the text or the figure caption.
- 6) Supplementary figure S4 – Could the authors specify which is the semiconducting hyperbolic metamaterial?
- 7) Main paper, Figure 3(a) and (b) – The authors compare the Purcell factor and propagation length in their 2H-TAS₂ HMM with that of an Ag/SiO₂ HMM and show that their Purcell factor and propagation lengths are much higher and as such, they perform better than the latter. However, one shortcoming I observe is that the range in which the hyperbolic dispersion is exhibited is much lower than that in the conventional Ag/SiO₂ HMM. While I understand that this is the case for just this particular material and one can get around this issue by building hetero structures made of different TMDs, I am curious as to why the authors use 2H-TAS₂ as their standard material. Why not use another material such as, say 1T-ZrTe₂ which is hyperbolic over a major part of the visible spectral range?
- 8) Main paper, Figure 3(d) – The authors compare the imaginary part of the dielectric function of

two different TMDs. While the values that they get are low, they are essentially the same as for a layered Ag/SiO₂ metamaterial. The authors have not shown the comparison of the imaginary part of the dielectric constant of their TMD compounds with that of the Ag/SiO₂ HMM, but a simple effective medium theory calculation that I performed showed that it is almost exactly the same. It would have been instructive to see a comparison of the imaginary part of the dielectric constants in these two systems too and a discussion on what brings about the much-improved Purcell factor in the TMD HMMs despite having the same amount of losses (and hence quenching) as the conventional HMM system. Is it just the fact that the system supports wave-vectors much larger than the conventional HMM system?

9) Main paper, Page 9, 1st paragraph, last sentence – The authors write, “The strong decay of the electromagnetic field inside the Ag layers leads to negligible hybridization between the surface plasmons implying that the broadband response is lost and explaining the narrow peak in the Purcell factor at 3.3 eV which originates from the unhybridized Ag surface plasmon.” Can the authors clarify what is implied by saying “the broadband response is lost”? The Ag/SiO₂ HMM has a broadband Purcell factor of 10⁴-10³ in the range 0-3.0 eV. So, to say that the broadband response is lost, does not seem true.

Since the paper is quite promising, if all the above concerns/questions/comments are addressed sufficiently, it can be considered for publication.

Reviewers' comments:

Reviewer #1 (Remarks to the Author):

Hyperbolic metamaterials has been considered as one of the most practical optical metamaterials and have drawn a lot of attentions. Recently, people found out there are natural existing crystals expressing hyperbolic dispersion, unless the optical loss is still quite high, especially at visible light frequencies. This manuscript studies the hyperbolic material properties based on layered transition metal dichalcogenides and shows the loss might be less than artificially structured metamaterials. The results seems to be very interesting. However, the reviewer do have a few questions regarding the modeling validity.

1. For the loss comparison between 2H-TaS₂ and Ag/SiO₂ multilayer in Fig. 3, the transmission peaks for these two kind of materials are different. The authors should find two systems have the same transmission peak and compare to each other.

We assume that the transmission peak the reviewer refers to must be must be the peaks in Fig. 3(b). This figure shows the propagation lengths through the hyperbolic medium and the reviewer request to substitute this comparison with a comparison between materials with high transmission peaks in similar spectral ranges. We are not sure this comparison would yield a meaningful result since the point of the comparison is to 1) show the wave-vector dependence of the hyperbolic dispersion for a conventional metamaterial and a natural hyperbolic material and 2) show the large propagation lengths that can be achieved. However, we partially accommodated the request by including a direct comparison of different natural hyperbolic materials (Fig. 3 in new manuscript) and a discussion of their differences.

2. It will be better if the authors can also compare the loss properties by including the case shown in Fig. 1(a).

The reviewer asks for a comparison between the propagation length in a metal nanowire medium also known as a type 1 hyperbolic metamaterial. We are unsure what such a comparison would achieve since it amounts to comparing a type 1 with a type 2 material which have different applications.

3. The optical properties of the heterostructures was calculated by the EMT (equation S3 and S4). This may not be correct if the material properties of each atomic layer components was just set to be the same as its bulky value. Does quantum effect play an important role here? This have to be explained in great details, perhaps in supporting materials.

The reviewer is correctly concerned with the validity of EMT for predicting the dielectric properties of van der Waals heterostructures. However, the validity of EMT was directly discussed in SI Section S3 for heterostructures of hexagonal boron nitride, graphene and tantalum disulphide where we found

EMT to be accurate for the prediction of hyperbolic regimes (Fig. S3) even for monolayer thin heterostructures. Furthermore, the validity of EMT for the calculation of the Purcell factor was discussed in detail in a paper by some of the authors (René Petersen et al. "Limitations of effective medium theory in multilayer graphite/hBN heterostructures", PRB 94, 035128 (2016)) where it was shown that the Purcell factor can be overestimated by EMT (Fig. 11 in said reference) but not enough to change our conclusions.

Reviewer #2 (Remarks to the Author):

While current manuscript considerably extends the number of known natural hyperbolic metamaterials, I do not find these findings novel enough. It is known for at least five years that several high T_c superconducting metamaterials are naturally hyperbolic. There was also a recent review of natural hyperbolic metamaterials published by Nature Photonics (E. E. Narimanov and A. V. Kildishev, "Metamaterials: Naturally hyperbolic", Nature Photonics 9, 214–216 (2015)). It is well understood that many materials having "layered" crystalline structure are hyperbolic. Thus, main finding of this manuscript is not new.

The reviewer is correct that natural hyperbolic materials are already known. We must however stress that the addition of the metallic TMDs to the class of layered materials is a novel and significant addition to the class of natural hyperbolic materials. Semiconducting natural hyperbolic materials are significantly limited in their applicability in that the hyperbolic properties are obtained due to interband transitions which leads to large intrinsic losses due to direct electron-hole excitations. The metallic TMDs have no direct transitions below the interband onset and losses are thus naturally low. Accordingly, we have made changes to the manuscript to emphasize the significance of the metallic natural hyperbolic materials.

The Authors also calculate Purcell factors for these materials using a well established formalism. However, they do not report any experimental data, which would confirm their findings. I cannot recommend this manuscript for publication in Nature Communications.

The authors have now included a comparison of the predicted hyperbolic regimes with experimental data (Fig. 2). This comparison shows that the theoretical predictions are qualitatively and semi-quantitatively in agreement with the experimental measurements. Quantitatively we estimate an uncertainty of approximately 0.5 eV for the hyperbolic regimes which will not change the conclusions of the study. Moreover, 0.5 eV is also the uncertainty of the experimental data judging from the (few) cases where more than one experimental dataset is available.

Reviewer #3 (Remarks to the Author):

The authors report the discovery of a class of non-magnetic transition metal dichalcogenide (TMD) compounds that have strong optical anisotropy exhibiting hyperbolic dispersion in the optical spectrum, using first-principles density functional theory (DFT) calculations. The authors then go on to calculate the Purcell factor for spontaneous emission enhancement in these compounds and show that it is much higher than that of a Ag/SiO₂ multilayered hyperbolic metamaterial (HMM). Then, they show that by combining different types of TMD HMMs in a layered fashion using van der Waal's forces, one can tailor further the range of hyperbolic dispersion.

Considering that conventional HMMs are limited by losses and lower propagation lengths, I find this

work, very intriguing and possibly as a fresh direction for the field of HMM.

However, I am of the opinion that it is not suitable for publication in a journal like Nature Communications at this stage.

The main reason is that it is fully theoretical and there is no comparison with any experimental findings to validate the results shown here. There is a recent publication [“Imaging spectroscopic ellipsometry of MoS₂,” J. Phys.: Condens. Matter 28(2016) 385301] that has shown ellipsometry measurements on MoS₂ (single and multilayer) and used an anisotropic model to describe its optical response. This paper, in fact, shows Type- II hyperbolic dispersion in the high energy (~3 eV) range. While, this is qualitatively similar to what the authors predict in this paper, it would considerably strengthen the results if they can also do a quantitative comparison of the dielectric constants provided in the above paper with those from their calculations and discuss the comparison of parameters such as Purcell factor and propagation length for the experimental and DFT dielectric constants. This would address some of the practical considerations too.

We have accommodated all the points above:

- We have now included a comparison of the predicted hyperbolic regimes with experimental data (Fig. 2). This comparison shows a good agreement with our predictions.
- We have also included a direct comparison between the calculated and experimental dielectric functions (supplementary figures 7-39) where data was available.
- The new Supplementary Figure 5 shows the “indicated” hyperbolic regimes based only upon the in-plane dielectric functions for which the number of experimental studies is considerably larger. This comparison further strengthens the validity of our results.
- The new Fig. 3 compares the Purcell factors and propagation lengths based on experimental and calculated dielectric functions for Graphite and 1T-ZrS₂ (where experimental data is available for both the in-plane and out-of-plane component of the dielectric tensor). The agreement with the experimental Purcell factors and propagation lengths is excellent.

Also, I suggest this publication should be cited by the authors as it has essentially demonstrated hyperbolic dispersion in MoS₂.

The cited experimental study provides optical constants on monolayer and tri-layer MoS₂ but these are not necessarily the same as for the bulk properties. However, we have included the provided reference in the paper within the discussion of the experiments:

“... measurements on three layers indicate a lower limit on the order of 2.9 eV[J. Phys.: Condens. Matter 28(2016) 385301].”

In addition to this, I have the following questions/comments:

1) Main paper, Page 4, paragraph 2 – The authors write: “As a unique feature, the metallic TMDs are hyperbolic below the onset of interband transitions....”. Does this imply that the hyperbolic dispersion feature arises from intraband transitions? **Yes, in fact, we regard this as a major result of the paper and have made changes in the content of the paper to emphasize its importance.**

If so, how is it that DFT calculations can take into account such transitions?

The intraband transitions are treated separately from the interband transitions. The intraband

contributions to the dielectric response is calculated as an integral of the Fermi-velocities along the Fermi surface. This essentially gives the in-plane and out-of-plane plasma-frequencies which causes the hyperbolic response, since the out-of-plane plasma-frequency is smaller than the in-plane plasma-frequency due to the small interlayer hybridization.

Secondly, if interband transitions have no role to play in the hyperbolic property, does it imply that the particular crystal structure of these materials is not a criterion in determining the hyperbolic dispersion behaviour?

There are a couple of points that must be raised to answer this question.

1. The crystal structure can still be crucial for the hyperbolic properties since it can determine whether the TMD is metallic or not. For example, due to different crystal field splittings 1T-MoS₂ is metallic while 2H-MoS₂ is semi-conducting.

2. Even if the interband transitions are not the direct cause of the hyperbolic properties of the layered material, they still play a role in the screening of the intraband response. The additional screening red-shifts the plasma frequencies and pushes the hyperbolic regime to lower frequencies.

2) As a follow up question, the authors refer to the TMDs as “metallic”. I am curious in what respect they are metallic? They have metal-like dielectric constants (with negative real part) in one of the directions (or one plane) over a certain frequency range. But, in the orthogonal direction, the dielectric constant is not metal-like. So, I am wondering how the TMDs can be referred to as metallic? Is this from the perspective of DC conductivity/resistivity where they show properties similar to those of metals?

The TMDs are metallic in the sense that they have a partially filled conduction band which gives a finite plasma-frequency and this definition is equivalent to the definition of a finite DC conductivity. However, the out-of-plane plasma frequency is typically much smaller than the in-plane plasma frequency. Thus the out-of-plane dielectric function is negative but only for small omega. The reviewer can verify this in the SI where the calculated dielectric functions have now been included. We have included our definition of a metal within the new manuscript.

3) Are there any practical difficulties in realizing such HMMs, for example, say, ensuring uniformity of behavior over a large lateral area (both with pristine TMDs and those from van der Waals heterostructuring)?

1. Due to the large amount of literature reporting the growth of single crystalline samples of the TMDs (some on the centimeter scale) we do not expect there to be any technical difficulties over a large lateral area. To emphasize this we have included the following sentence in our study: “Finally, with studies reporting growth on the centimeter scale [Greenaway1965, hill1972cg], we do not expect any technical difficulties in the experimental synthesis high-quality materials with large lateral areas. “

2. The production of vdW heterostructures is still very much in its infancy but many studies report production of large-area heterostructures on the scale of tens of microns. We have emphasized this point in the paper by including references to studies that produce large area heterostructures: “The practical feasibility of the proposed large scale nanostructured heterostructures is supported by an increasing amount of experimental studies showing practical realizations of high quality large area

heterostructures[tongay2014,Zhang2015,Wu2015,Xeno2015,Samad2016]”

4) Supplementary section S2 – What was the value for the electron relaxation rate used in the calculations of the Drude response (equations S1 and S2)? How was this determined?

We realize that we were not completely clear on this point. The electron relaxation rate was included according to a new model of the electron relaxation rate designed to take into account effects of the bandstructure of the materials which was described in detail in ref. “Gjerding, M. N., Pandey, M. & Thygesen, K. S. Band structure engineered metals for low-loss plasmonics. Preprint, arXiv (2016)” recently accepted in Nature comm. and pending publication. We have stressed this in the new manuscript. In summary, the relaxation rate is $\eta \propto \text{JDOS}(\omega) / \omega$ where the proportionality constant is the same for all materials. It has been fixed to correspond to a constant relaxation rate in silver of approximately 30 meV. The relaxation rate employed in the Drude response of the metallic compounds is also presented in supplementary figures 7-39 (only shown for metallic compounds).

5) Supplementary figure S2(a) – What are the two different cases (dotted and solid lines) of dielectric constants plotted? This does not seem to be mentioned clearly in the text or the figure caption.

Solid lines are the real parts of the dielectric function and dashed lines are the imaginary parts. To clarify we have included the following sentence in the caption:

“Real part of the dielectric functions are shown by full lines (blue: in-plane component, green: out-of-plane component) and dashed line show the imaginary parts (red: in-plane component, violet: out-of-plane component)”

6) Supplementary figure S4 – Could the authors specify which is the semiconducting hyperbolic metamaterial?

We have now indicated the semi-conducting material in Sec. S4 by changing the following sentence: “Only one semi-conducting hyperbolic material has been included (1T-ZrS₂) since all of them behave similarly above their interband onset”

7) Main paper, Figure 3(a) and (b) – The authors compare the Purcell factor and propagation length in their 2H-TAS₂ HMM with that of an Ag/SiO₂ HMM and show that their Purcell factor and propagation lengths are much higher and as such, they perform better than the latter. However, one shortcoming I observe is that the range in which the hyperbolic dispersion is exhibited is much lower than that in the conventional Ag/SiO₂ HMM. While I understand that this is the case for just this particular material and one can get around this issue by building hetero structures made of different TMDs, I am curious as to why the authors use 2H-TAS₂ as their standard material. Why not use another material such as, say 1T-ZrTe₂ which is hyperbolic over a major part of the visible spectral range?

First, we realize that we have not done a good job of bringing forth the point of the importance of the metallic TMDs. We have changed the manuscript accordingly to emphasize their importance. To answer your questions:

We have now included a new figure (Fig. 3) which compares the Purcell factors and propagation lengths of 2H-TaS₂, 1T-ZrS₂ (theory and exp.) and graphite (theory and exp.). This figure shows clearly the advantageous properties of the metallic TMDs compared to semi-conductors. 2H-TaS₂ is

particularly interesting since it exhibits simultaneously large Purcell factors and low losses in contrast to any semi-conducting natural hyperbolic material. In particular, we expect the metallic TMDs to be particularly interesting at the telecom wavelength (0.8 eV \sim 1.55 micron) where the propagation length is long and the Purcell factor is high. We also compare the semi-conducting 1T-ZrS₂ (which is hyperbolic in a part of the visible regime) with the Silver-SiO₂ metamaterial in Supplementary Figure 4 and we do indeed find that the semiconducting TMDs have larger Purcell factors in the visible range.

Another point of the Figure is to show that even if EMT predicts hyperbolic properties of the Ag/SiO₂ metamaterial it is, in fact, not hyperbolic due to the small overlap between surface plasmons on the Ag/SiO₂ interfaces. We have changed the text accordingly to emphasize the loss of the hyperbolic response.

8) Main paper, Figure 3(d) – The authors compare the imaginary part of the dielectric function of two different TMDs. While the values that they get are low, they are essentially the same as for a layered Ag/SiO₂ metamaterial. The authors have not shown the comparison of the imaginary part of the dielectric constant of their TMD compounds with that of the Ag/SiO₂ HMM, but a simple effective medium theory calculation that I performed showed that it is almost exactly the same. It would have been instructive to see a comparison of the imaginary part of the dielectric constants in these two systems too and a discussion on what brings about the much-improved Purcell factor in the TMD HMMs despite having the same amount of losses (and hence quenching) as the conventional HMM system. Is it just the fact that the system supports wave-vectors much larger than the conventional HMM system?

Yes, the huge Purcell factor enhancement essentially originates from the support of large wave-vectors compared to the Ag/SiO₂ metamaterial. We hope with the changes to the manuscript that the origin of the large Purcell factors are more clear.

Additionally, an important point is that the effective medium theory for the dielectric function is simply not valid for the Ag/SiO₂ metamaterial. Such a metamaterial cannot be described by the simple averaging procedure of EMT due to the large metallic screening of silver.

9) Main paper, Page 9, 1st paragraph, last sentence – The authors write, “The strong decay of the electromagnetic field inside the Ag layers leads to negligible hybridization between the surface plasmons implying that the broadband response is lost and explaining the narrow peak in the Purcell factor at 3.3 eV which originates from the unhybridized Ag surface plasmon.” Can the authors clarify what is implied by saying “the broadband response is lost”?

To clarify we have changed the sentence to: “the ~~broadband~~ hyperbolic response is lost”.

The Ag/SiO₂ HMM has a broadband Purcell factor of 10^4 - 10^3 in the range 0-3.0 eV. So, to say that the broadband response is lost, does not seem true.

We think this question is related to the distinction between being “broadband hyperbolic” and having a “broadband large purcell factor”. To clarify we have now included in the new Supplementary Figure 6 the calculated Purcell factor and propagation length using both the transfer matrix method (TMM) and effective medium theory (EMT) for the Ag/SiO₂ metamaterial. Comparing these methods show that the Purcell factor enhancement predicted by EMT is much greater than the prediction of the TMM. EMT shows a clear shoulder at 2.8 eV where the Ag/SiO₂ become hyperbolic according to EMT, however, the TMM method shows no such signature. The reason is the strong dielectric screening of the metal

(not taken into account by EMT) which inhibits the propagation of surface plasmons through HMM and the resulting Purcell enhancement is due to surface plasmon polaritons and quenching, i.e., it is not due to hyperbolic modes.

In summary: The broadband response is not due to hyperbolic modes but quenching and single-interface surface plasmon polaritons.

Since the paper is quite promising, if all the above concerns/questions/comments are addressed sufficiently, it can be considered for publication.

Reviewers' Comments:

Reviewer #2:

Remarks to the Author:

My opinion on this manuscript has not changed after the revision. This manuscript is quite comprehensive. However, in my opinion it is not novel enough to justify publication in Nature Communications.